# Four Futures for Occupational Safety and Health

**DOI:** 10.3390/ijerph20054333

**Published:** 2023-02-28

**Authors:** Sarah A. Felknor, Jessica M. K. Streit, Nicole T. Edwards, John Howard

**Affiliations:** 1Office of the Director, National Institute for Occupational Safety and Health, Atlanta, GA 30333, USA; 2Office of the Director, National Institute for Occupational Safety and Health, Cincinnati, OH 45226, USA; 3Office of the Director, National Institute for Occupational Safety and Health, Morgantown, WV 26505, USA; 4Office of the Director, National Institute for Occupational Safety and Health, Washington, DC 20024, USA

**Keywords:** strategic foresight, occupational safety and health, scenarios, alternative futures, drivers of change, data security, mental health, partnerships, virtual work

## Abstract

Rapid changes to the nature of work have challenged the capacity of existing occupational safety and health (OSH) systems to ensure safe and productive workplaces. An effective response will require an expanded focus that includes new tools for anticipating and preparing for an uncertain future. Researchers at the U.S. National Institute for Occupational Safety and Health (NIOSH) have adopted the practice of strategic foresight to structure inquiry into how the future will impact OSH. Rooted in futures studies and strategic management, foresight creates well-researched and informed future scenarios that help organizations better prepare for potential challenges and take advantage of new opportunities. This paper summarizes the inaugural NIOSH strategic foresight project, which sought to promote institutional capacity in applied foresight while exploring the future of OSH research and practice activities. With multidisciplinary teams of subject matter experts at NIOSH, we undertook extensive exploration and information synthesis to inform the development of four alternative future scenarios for OSH. We describe the methods we developed to craft these futures and discuss their implications for OSH, including strategic responses that can serve as the basis for an action-oriented roadmap toward a preferred future.

## 1. Introduction

There is evidence that rapid and multifaceted social, technological, environmental, economic, and political (STEEP) changes have noteworthy and complex effects on the nature of work, the workforce, and the workplace [1,2,3,4,5]. These changes have had a demonstrable impact on the practice of occupational safety and health (OSH), and these trends are expected to continue [6,7,8,9]. It has been argued that an expanded focus for OSH will be necessary to proactively prepare for, and respond to, these changes [10]. This includes broadening the range of factors that are recognized as affecting workers and the type of outcomes we consider relevant to OSH [9,10]. The need for expanding paradigms to anticipate and prepare for the changing conditions of OSH has been reported and calls for new strategic approaches to support the transition from OSH 4.0 to OSH 5.0 [11]. Previous work has also substantiated the value of scenarios to identify potential new and exacerbated hazards in the future of work [12].

These conceptual expansions of OSH will require new approaches to research and practice designed to protect and promote the future of worker safety, health, and well-being. In a previous publication, we proposed strategic foresight as an innovative and systems-focused method well-positioned to support the expanded OSH paradigm [13]. Strategic foresight is an action-oriented planning discipline grounded in futures studies and strategic management. It uses established techniques and methods to create well-informed future scenarios that help organizations better prepare for potential threats and take advantage of new opportunities [13,14]. Scenarios are evidence-based stories about how the future might be different. They share findings in a way that personalizes data often presented in charts or policy briefs [15,16,17]. They help us experience emerging trends and drivers so we can better imagine how these forces might interact and shape the future. It is important to note that the scenarios produced by strategic foresight are not intended to accurately forecast the future or definitively answer questions about which imagined future will unfold. Instead, they are meant to explore a wide range of opportunities and challenges associated with possible futures that could arise [18,19]. 

While foresight does not predict the future, its contributions to strategic planning can help avoid surprise and identify pathways to preferred future outcomes [13,19,20,21]. A number of world-renowned corporations, such as Royal Dutch Shell (now Shell plc), General Electric, Siemens, and Daimler AG, have used strategic foresight and scenarios to improve their resilience and performance during periods of intense challenge and rapid change [22,23]. Recently, strategic foresight and scenario-based planning have also gained favor within the U.S. federal government. Many agencies, including the U.S. National Institute for Occupational Safety and Health (NIOSH), are now applying foresight methods to enhance the future readiness of their research programs and practice activities [24,25].

This paper describes the inaugural NIOSH strategic foresight project, designed to promote institutional capacity in strategic foresight and improve preparedness by considering how the future will impact occupational safety and health (OSH) research and service activities. The activities described in this paper were carried out by a multidisciplinary team of NIOSH scientists and analysts (Appendix B) who represented 10 of 12 divisions, laboratories, and offices at NIOSH. This team, led by co-authors SF, JS, and NE, performed all activities and analyses described in this paper. 

## 2. Materials and Methods

### 2.1. Foresight Framework for Occupational Safety and Health

We adapted the scenario-based Framework Foresight developed by University of Houston to consider alternative futures for occupational safety and health [13,26]. The framework (Figure 1) provides a stepwise series of activities that develop complementary and plausible futures based on research inputs, which are analyzed to identify strategic implications and issues that should be considered as part of strategic planning and decision-making for the future.

### 2.2. Framing the Domain

The first step of the foresight framework involves framing the domain or topic of interest (Figure 1. Stage 1). The domain frame establishes the scope of the project, defines the relevant parameters of the main topic of interest, and assesses the current conditions of the domain to help distinguish the future from the present.

#### 2.2.1. Domain Map

Our project was designed to answer the key question: How will the future impact NIOSH research and service activities? The main domain, or subject of the project, was the future of occupational safety and health in the U.S. The domain map, presented in Figure 2, guided our search for strong and weak signals of change that might affect the future of OSH in the primary and secondary domain topics. The primary domain topics were defined as follows: Facilities, Policies, OSH Workforce, Resources, OSH Activities, and STEEP. Secondary domain topics were identified for each primary topic to further refine the search for information. Facilities included consideration of laboratories, offices, and equipment. Policies considered information technology, human resources, and science policy. OSH Workforce considered issues of supply and demand, research and non-research workforce, and workforce development. Resources considered data, access to research settings, study populations, and partners. OSH Activities considered OSH research and service. STEEP included consideration of the social, technological, economic, environmental, and political environment. With NIOSH as the focus of this exercise, the scope was necessarily limited to experiences in the U.S.

#### 2.2.2. Time Horizons

We used the Three Horizons Framework (Figure 3) to connect the present to the future by thinking about current assumptions, emerging changes, and possible desired futures [27,28]. The Three Horizons Foresight method gives OSH a tool for thinking about how change occurs by considering that three qualities of the future are visible to one degree or another in the present [13,28]. Horizon 1 (H1), the current way of doing things, is unlikely to be the most appropriate or effective course of action as change occurs over time and the level of strategic fit within the environment declines. Instead, what appear as ‘fringe’ or ‘marginal’ ideas in the present eventually become the new way of doing things in the far-term future (Horizon 3; H3). Situated in between is a mid-term future (Horizon 2; H2) characterized by high levels of instability and change as we transition out of H1 and into H3.

In the context of a strategic foresight project, time horizons are often tied to the normal business cycles of an organization [28,29]. For this project, we aligned the time horizons with NIOSH strategic planning cycles. H1, the short-term future, was defined as the period 2021–2026; H2, the mid-term future, was defined as the period 2027–2036; and H3, the long-term future, was defined as 2037 and beyond. This approach provided a framework to consider the short-term, mid-term, and long-term future of OSH in the U.S.

#### 2.2.3. Current Assessment

Before starting our research on the future, we considered the key factors influencing the domain in the present. We developed a current assessment of the conditions, key interest groups, and recent history to provide some context for our inquiry into the future. This is not intended to be a comprehensive review of the current state of OSH. The current assessment is the culmination of input provided by team members and senior NIOSH leaders, who represent the field of OSH dating back to the inception of NIOSH in 1970. We organized this assessment, presented in Table 1, Table 2 and Table 3, around the primary categories shown in the domain map (Figure 2). Summaries of our findings are supported with links to references containing more information. The Appendix A provide a glossary of acronyms used in these tables and throughout the paper (Appendix A).

## 3. Research

With the current assessment in place, we turned our attention to the future. Our research on the future started with scanning for information on early signals of change. Scanning (Figure 1. Stage 2) involves reviewing a variety of sources for information about how things might be different in the future. While we included mainstream peer-reviewed literature, we also had to move outside the prevailing literature search paradigms to ‘find the fringe’, which consists of unconventional sources providing (often anecdotal) evidence of weak or early signals of change [53]. In futures work, strong evidence confirms the baseline or what we already know. The goal of scanning is to find the weak signals, focus on what is *different* from the baseline, and look across all time horizons [20,54]. These signals may come from a variety of sources, such as publicly available reports, blogs, internet searches, social media, and other information on the domain of interest [55,56]. 

### 3.1. Scanning

We found and cataloged over 240 relevant scanning hits that provided information across all three time horizons and in each of the primary and secondary categories from the domain map (Figure 2). The scanning hits became the library for this project. 

We synthesized and coded the scanning hits according to multiple criteria to provide further insights into the direction, specificity, timing, plausibility, and NIOSH preparedness to respond to the potential change. To organize the scanning library, each scanning hit was also labeled as a trend, issue, plan, or projection about how the future might be different. *Trends* describe specific quantities of change moving in one direction or another. *Issues* represent current, emerging, or potential conflicts, controversies, dilemmas, or choices not yet made that will influence the trajectory of the future. *Plans* are the published intentions of key interest groups to create change in the future, and *Projections* are publicly announced forecasts that might influence the future. Collectively known as TIPPS, this organized set of scanning data is one of the main outputs of strategic foresight research, and it is used to inform subsequent project stages [26]. To ensure the volume of data we carried forward remained manageable, we used the TIPPS framework to condense our scanning hits based on the content overlap. For example, all scanning ‘trends’ describing the evolution of OSH toward a more holistic model of worker well-being were combined into a single trend statement, and all ‘issues’ related to gig work were consolidated into a single question that articulated the underlying issue. This data reduction exercise resulted in a final set of 119 TIPPS for the current project, which is provided as a categorized list in the Appendix A [6,57,58,59,60,61,62,63,64,65,66,67,68,69,70,71,72,73,74,75,76,77,78,79,80,81,82,83,84,85,86,87,88,89,90,91,92,93,94,95,96,97,98,99,100,101,102,103,104,105,106,107,108,109,110,111,112,113,114,115,116,117,118,119,120,121,122,123,124,125,126,127,128,129,130,131,132,133,134,135,136,137,138,139,140,141,142,143,144,145]. 

### 3.2. Drivers of Change

The next stage of our project involved identifying drivers of change. Drivers are essential elements of all scenarios, and they are synthesized from thematic clusters of scanning results (e.g., the TIPPS). They reveal key evidence-based developments from today that are likely to affect or shape the future [26]. Eight key drivers emerged from the synthesis of our research. Brief descriptions of the drivers, which are the big changes underway in a domain that can be projected to elucidate how the future might be different from today, are provided in Table 4.

### 3.3. Cross-Impact Matrix

With the drivers defined, we constructed a cross-impact matrix, an analytical technique used in foresight to explore how drivers might interact in the future [26,146,147,148,149,150]. Some drivers may influence only one element of the system, while others might influence several elements. This exercise, which was completed by the foresight team using a consensus approach to evaluate the relative impact of drivers, helps to identify neutral drivers, which have a neutral impact across the matrix while being reinforced or contradicted by other drivers and should therefore be considered for removal before moving to the scenario building phase of the project [148,149,150].

The results of our eight-driver cross-impact matrix are shown in Figure 4. The matrix should be interpreted, “as Driver X (independent driver) occurs, the impact on Driver Y (dependent driver) is…” one of the following outcomes: strongly reinforcing, reinforcing, neutral, contradictory, or strongly contradictory. We used a 5-point scoring scheme for the different relationship outcomes, where 5 = strongly reinforcing impact, 4 = reinforcing impact, 3 = neutral impact, 2 = contradictory impact, and 1 = strongly contradictory impact. 

Our assessment provided no evidence of neutral drivers. All eight drivers had sufficient impact to continue into the scenario-building process. Of note, Advanced Technologies and the Virtual Workplace had the strongest reinforcing score of 33, suggesting they would likely play a key role in any future we develop. 

## 4. Results

For this project, we used four generic futures, or archetypes, as the blueprints for developing our scenarios. The archetypes are based on decades of cross-cultural investigation conducted by the Hawaii Research Center for Futures Studies under the direction of Dr. Jim Dator and adapted by the University of Houston for a wider range of project work [151,152]. The results of this work concluded that images of the future that exist across the world could be classified into one of four broad groups. Descriptions and critical development questions for each archetype are provided in Figure 5.

To move to the next step in developing the scenarios (Figure 1. Stage 3), our full project team divided into four groups to craft the initial draft of a 400- to 600-word future scenario according to their assigned archetype. To fully explore the future scenario, the groups were encouraged to carefully consider all eight drivers during the drafting process, selecting those that seemed the most influential in shaping their assigned scenario. Each driver had a different story to tell as it played out across the four futures, and all drivers had a role in at least one of the scenarios. A description of each driver in their archetypal future is provided in the driver map in the Appendix A), which allows for a direct comparison of the role of each driver across the four futures. The group drafts were subsequently compared to ensure they told four unique stories of the future, and the writing styles and tones of the stories were harmonized by the authors of this paper to improve their cohesion as a complementary set of future views. The title, abstract, and key drivers for each of the final four scenarios are provided in Section 4.1.

### 4.1. Future of OSH Scenarios

Table 5 summarizes the four futures of OSH scenarios by their archetype and provides a brief description of the key characteristics of each. 

#### 4.1.1. Boundaries Continue to Blur

In the Continuation future, boundaries related to work locations, employment arrangements, work hours, the interface between work life and personal life, and the human–machine interaction continue to blur. Automation and virtual capabilities continue to alter how work is performed. Advances in technology introduce new hazards into the traditional work environment while also exacerbating existing risks, and the line between work and personal life continues to blur as more work is done remotely. Data collection and machine learning methods make processes more efficient, but they are also increasingly used to monitor, measure, and evaluate workers. Nonstandard work arrangements are more mainstream across industries, which complicates the management of safety and health. Economic trends, individual preferences, and competing priorities are extending the working-life continuum, and workers find it challenging to keep pace with technological advances through upskilling and reskilling. 

Key drivers influencing the Continuation scenario are 

Virtual Workplace: The boundaries between work and personal life are further blurred with advanced communication and data channels. Workplaces, defined by function, are ubiquitous, and OSH is challenged to protect workers in virtual environments.Nonstandard Work Arrangements: Nonstandard work arrangements become more mainstream, and employers embrace hybrid environments that favor the individual over the collective.Workforce: Economic trends, individual preferences, and competing priorities result in delayed retirement and an extended working-life continuum. Workers seek to upskill or reskill to stay competitive in the workforce.Advanced Technology: Advances in technology and AI dramatically increase productivity. Reliance on these new technologies outpaces the rates at which workers can be retrained, and OSH systems can be built to address hazards created by these new technologies

#### 4.1.2. The Perfect Storm

The Collapse future is a perfect storm, where failure to adapt coupled with a lack of trust and resources forces people and organizations to rely on themselves, to the detriment of worker health and safety. The OSH community no longer meets worker needs and cannot adequately protect and promote worker safety and health. Businesses and workers are in survival of the fittest mode, stemming from a polarization of society that has significantly weakened public services (including OSH research). Lack of trust in government due to mis- and dis-information, fear of being monitored, and low data transparency limit opportunities for research and practice. There are insurmountable challenges for communicating scientific evidence to improve working conditions.

The OSH system weakens as it splits to address concurrent demands for safe and healthy work in both virtual and physical work settings. Demand for improved technology far outpaces the supply, making virtual work unaffordable. Distrust in AI and electronic surveillance leads to communities of people living with limited use of technology. The utopia of work flexibility has not occurred, as the growing complexity of unstructured and fluid work arrangements becomes unsustainable. Automation and robotics replace mid-skill jobs, leading to chronic underemployment/unemployment. The result is a population with increasing mental and physical health issues.

Key drivers influencing the Collapse scenario are 

Data Security: Failure to install adequate online database protection results in hacking of personally identifiable information (PII) and biometric data and a loss of expert credibility. Lack of trust, fear of being monitored, and low data transparency limit opportunities for OSH research and practice.Nonstandard Work Arrangements: The growing complexity of unstructured and fluid work arrangements eventually becomes unsustainable. As a result, workplaces shut down, forcing “survival of the fittest” for organizations and workers.Advanced Technology: Automation and robotics replace all manual jobs, leaving fewer opportunities for work. Distrust in AI and electronic surveillance leads to communities of people living with limited use of technology.Virtual Workplace: Demand for improved technology far outpaces the supply, making virtual work largely unaffordable. The OSH system weakens as it bifurcates to address concurrent demands for safe and healthy work in both virtual and physical work settings.Knowledge Generation: Mis- and dis-information abound as trust in government further deteriorates, and authority decentralizes. This creates insurmountable challenges for communicating scientific evidence to improve working conditions.

#### 4.1.3. Remote Controlled

In the New Equilibrium future, demands for new research on worker-centric arrangements, remote work, and human–machine collaborations strongly influence the allocation of OSH resources. Post-pandemic, many organizations adopt a hybrid operations model. Enhanced data security facilitates remote work for most computer-based duties, and workers manage their own portable benefits portfolio as they frequently change jobs. Though employers feel pressure to demonstrate social responsibility to attract and retain top talent, the implementation of health-supporting initiatives is challenged by the highly remote, highly diverse workforce. Consequently, workers find themselves primarily responsible for managing their own long-term exposures, chronic health conditions, and work-life integration. Advanced technology is embraced across industries, but energy use regulations and notable workforce skills gaps slow the rate of implementation. However, reskilling and upskilling opportunities are limited as organizations focus on maximizing outputs through optimized human–machine collaboration. Amid these changes, OSH priorities and resource allocation center on developing energy-efficient data collection and monitoring techniques, evaluating the ‘remote workability’ of people and jobs and linking disparate data systems to understand worker-centric employment arrangements.

Key drivers influencing the New Equilibrium scenario are 

Virtual Workplace: Employers pare down central facilities as they adopt remote work practices. OSH becomes responsible for developing valid assessments of ‘remote workability’ for jobs and workers.Nonstandard Work Arrangements: Workers manage their own careers, benefit portfolios, and risks as they frequently move between nonstandard employers. OSH struggles to study and manage hazardous exposures.Data Security: Increased investments support the development of effective built-in data security systems. While users must adapt to changing conditions, they largely gain through improved data security, linkages, and efficiency.Climate and Energy: The focus shifts from developing alternative energies to limiting energy use, which impacts work in the energy sector and presents new challenges for OSH innovation and efficacy.Advanced Technology: The implementation of advanced tech in the workplace is constrained by energy use regulations and notable workforce skills gaps. Organizations focus on human-machine collaboration to maximize outputs.

#### 4.1.4. One World Health

The transformation future is an advanced tech world, mental health, and data protections become central elements of an expanded OSH paradigm, research is driven by population need, and industries achieve one world health to sustain global workforce well-being. Most work is done remotely, and industry sectors evolve as goods and services are created and delivered virtually. The workforce is more distributed and flexible, with multiple and global employers. AI and robotics combine to create unparalleled capacity, and human labor is only required for social and artisanal tasks or coordinating AI in novel ways. The OSH paradigm expands to address worker mental health issues exacerbated by advances in technology and AI. Data and data systems become integrated with individual workers’ identities and their ability to function. Protecting worker data is as important as ensuring worker safety and health. Through technological augmentation, workers become more capable, long-lived, and less prone to maladies. Workers unable to adjust to these changes depend on Universal Basic Income. Social credit scores are a new incentive structure for OSH, and independent entities create searchable and verifiable OSH Quality (OSHQ) scores. Total worker health and total environment health are integrated into One World Health, where the global workforce and the planet thrive.

Key drivers influencing the Transformation scenario are 

Social Credit: Social credit scores are a new incentive structure for OSH. Independent credentialing entities create searchable and verifiable OSHQ scores, and companies with high OSHQ scores are sought after by investors and workers.Workforce: The workforce is more distributed and flexible with multiple and global employers. OSH has expanded to address worker mental health challenges associated with the complexity of interfacing with advanced technologies. Workers unable to adjust to these changes depend on Universal Basic Income.Data Security: Data and data systems become integrated with individual workers’ identities and their ability to function. Protecting worker data is as important as ensuring worker safety and health.Climate and Energy: Total worker/total environment health is expected.Advanced Technology: Improving AI and robotics combine to create unparalleled capacity in analysis and assembly, and human labor is only required for social and artisanal tasks or coordinating AIs in a novel way. Cybernetics, advanced biotechnology, and genetic engineering can augment workers to be more capable, long-lived, and less prone to maladies.

The scenarios were designed with their utility for future planning and action in mind. They were grounded in the initial mapping of drivers across each of the four archetypes shown in the Appendix A. The driver map was a useful way to keep the four scenarios discrete and avoid overlap. The continuation scenario established the baseline for the future of OSH along a linear trajectory. The collapse scenario, while the more dystopic scenario, provided a roadmap of future issues to avoid, and the new equilibrium scenario offered a path forward from the current baseline. Finally, the transformation scenario described a future that conveys aspirational goals for the future of OSH.

### 4.2. Key Strategic Issues

The last phase of our analysis identified key strategic issues that represent fundamental changes or challenges embedded within the scenarios (Figure 1. Stage 4). We broadly considered the implications and impacts these would have on the OSH system across H1 (2021–2026), H2 (2027–2036), and H3 (2037 and beyond). A thematic analysis was conducted by the foresight team leads (JS and SF) to identify the underlying strategic issues, which were vetted with and agreed upon by the foresight team. The strategic issues were organized into the five overarching strategic focus areas shown in Table 6: data security, mental health, partnerships, research, and virtual work.

As we considered how to connect key strategic issues to action, we used the original horizon timeframe (H1, H2, H3) to link issues to recommendations for the near, mid, and far term, recognizing that the time reference is not an absolute marker for when action should be considered or taken. The issues of data security, the mental health of the workforce, and the future OSH research portfolio loomed large across all scenarios in terms of their potential impact (positive or negative) and the actions that will be needed to address these issues.

## 5. Recommendations

Recommendations for strategic options and actions (Figure 1. Stage 5) were derived from the key strategic issues and focal areas described above. Our recommendations follow a phased approach, which provides a time-oriented roadmap to the future. The sequence described in Table 7 should be considered a loose guideline.

The near, mid, and far-term structure provided a general sense of when strategic actions might be needed to address emerging issues. However, responsive strategic actions can and should occur at any juncture based on opportunity and need. A robust strategic response allows an organization to identify strategic issues across multiple scenarios to inform strategic decisions that provide an integrated response to multiple conditions of uncertainty.

## 6. Conclusions

This project produced four well-informed scenarios of plausible alternative futures for OSH, providing insight into the major trends and inputs driving or shaping the future. The key strategic issues may challenge preconceived notions, which can be useful in moving OSH organizations from a reactive to a proactive approach to thinking about and planning for the future [154]. Rather than thinking of the future as a straight line from the past, an operational framework that includes different views of alternative futures that may result from system disruptions, innovations, and other uncertainties can create a more future-focused organization with more robust and resilient policies and programs [17,155]. 

Connecting the work of strategic foresight to organizational planning and action is a critical step that requires institutional commitment and a futures orientation. The long-standing success stories from the corporate sector that connects strategic foresight to planning and action provide useful but limited guidance for federal government agencies. NIOSH is currently exploring different approaches to strategic action that best align with internal planning cycles and capacity. One approach develops strategic actions that are scenario-specific or time horizon-specific based on different and distinct futures. This approach requires an agile planning paradigm that can manage simultaneous potential strategic actions. Another approach develops an integrated set of strategic options and actions based on an assessment of issues that would have the greatest impact if they occurred and for which the organization is the least prepared. This second approach allows for planning and action resources to be focused on preparing the agency for different possible futures based on a prioritized set of key strategic issues. It is to be determined which approaches, or perhaps a hybrid will be most useful in a federal agency over time.

The final stage of the Foresight Framework for OSH calls for continual monitoring of the domain of interest (Figure 1. Stage 6) to look for new signals of change that might influence the future. That activity is ongoing and will inform future projects and scenarios. 

While this work was limited in scope to OSH in the U.S., the methods and approach used to apply strategic foresight to this field may have value outside of the geographic domain of this project. 

## Figures and Tables

**Figure 1 ijerph-20-04333-f001:**
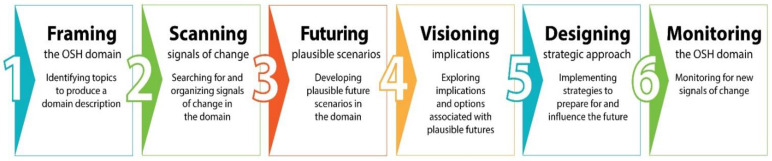
Foresight Framework for OSH, first published in Streit et al. (2021) [13]. While the framework is presented as a sequential model, the process is not entirely linear. The end of each stage should consider work completed in previous stages to determine if additional work is needed before moving on to the next stage. During each stage of the current project, we critically reviewed the outputs from the previous stages to determine if additional work was needed before proceeding, creating a sequence much like a feedback loop in a logic model. Our application of this approach, described in detail throughout the rest of the paper, provided a solid foundation for moving through the full strategic foresight process.

**Figure 2 ijerph-20-04333-f002:**
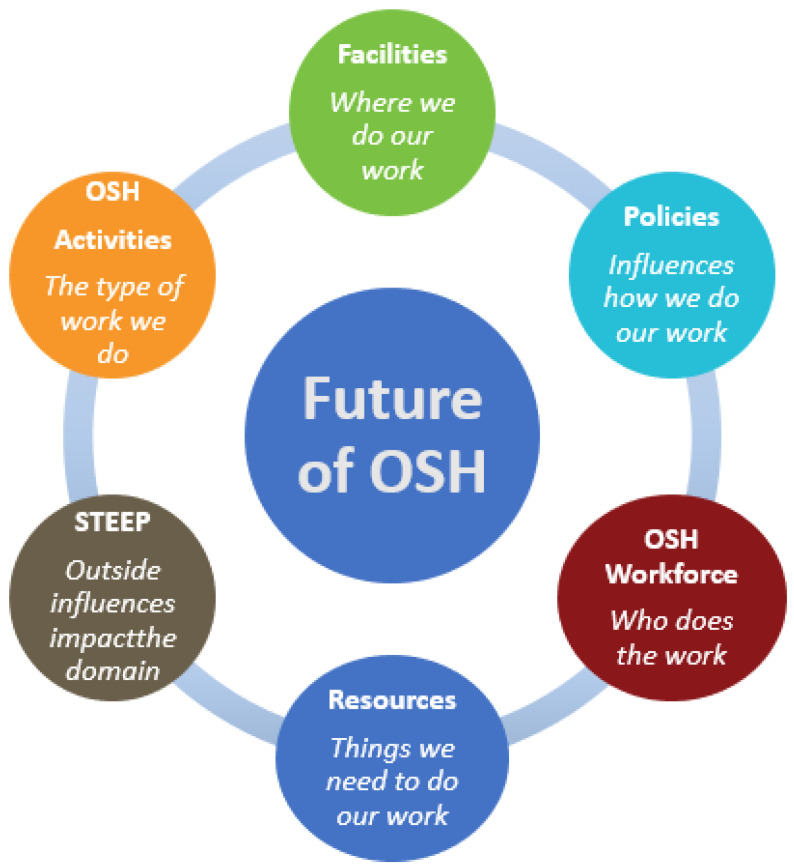
Doman map for the Future of OSH.

**Figure 3 ijerph-20-04333-f003:**
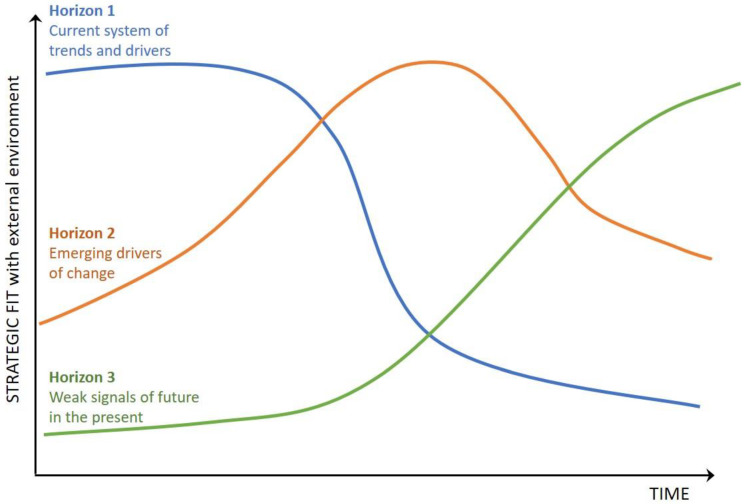
Visual representation of Three-Horizon Foresight [28], first published in Streit et al. (2021) [13].

**Figure 4 ijerph-20-04333-f004:**
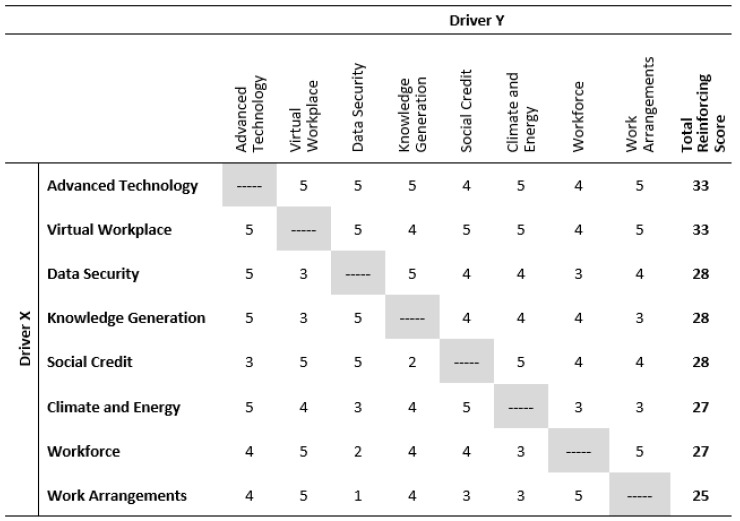
Cross-impact matrix for drivers of change.

**Figure 5 ijerph-20-04333-f005:**
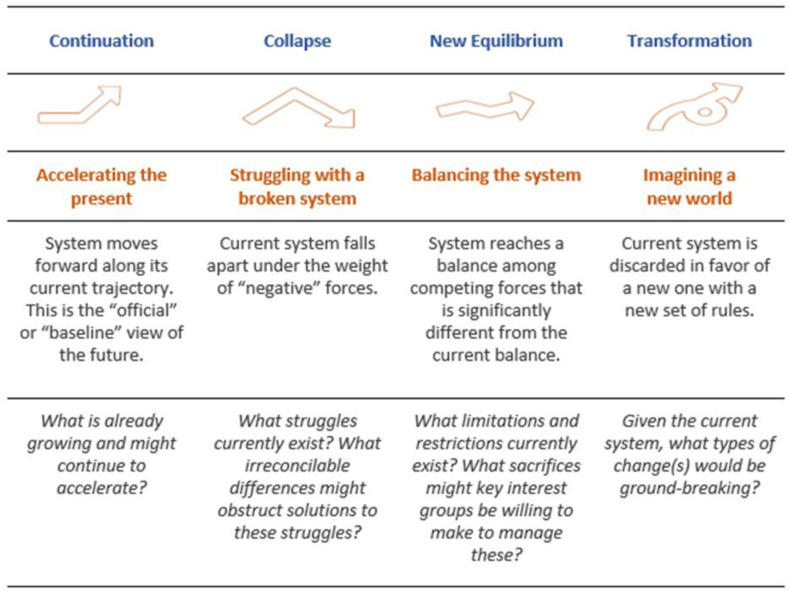
Authors’ overview of four archetypes, informed by guidance materials from Hines & Bishop (2015) [26], Dator (1979, 2009) [151,152], IFTF (2021) [153].

**Table 1 ijerph-20-04333-t001:** Current conditions.

Category	Description
Activities	Research is driven by the NIOSH Strategic Plan and the burden, need, and impact (BNI) framework [30,31]. Service activities rely on key partnerships to address a variety of mandates and stakeholder needs. Employee wellness is recognized as integral to OSH and important to bottom line [6].
Facilities	NIOSH maintains geographically dispersed research facilities with varying states of capabilities, enhanced by recent significant investment in improvements (e.g., Cincinnati, OH facilities) [32]. Access to facilities was limited during COVID-19 due to safety protocols [33]. CDC and NIOSH promote the design and implementation of healthy remote work options [34].
Policies	Policies to address privacy and security issues affect how data are consumed and interpreted [35,36,37,38]. There is increased demand for diversity, equity, inclusion, and accessibility; evolving definitions and conditions of employment; and an increasing focus on psychosocial health and well-being [39,40,41].
Resources	Data systems and database linkages are evolving with technological advances, which brings increased security concerns. Human subject research programs and access to U.S worksites were suspended during COVID-19. Access to study populations is governed by federal human research protection program policies [42]. Strategic partnerships with key interest groups are critical to NIOSH work but can be challenging to establish and maintain.
STEEP	The rapid and evolving nature of STEEP directly influences research and service activities and impacts key interest groups and funding appropriations.
Workforce	NIOSH has an aging workforce (25% of staff are currently retirement eligible; 40% will be eligible in the next five years). The multigenerational working environment requires continuous adaptability aimed toward collaboration, training, and productivity [43].

**Table 2 ijerph-20-04333-t002:** Key interest groups.

Category	Description
Academia	University-based agencies, both NIOSH grantees and non-grantees, who support current research efforts and identify integration and implementation opportunities
Employer Organizations	Individuals/groups of employers and trade associations who partner to conduct research and provide findings and interventions to change the workplace and are instrumental in some research and service activities
Federal Advisory Committees	Provide formal and informal guidance, recommendations, and input to the Institute
Federal Partner Agencies	EPA, OSHA, MSHA, NIH, NIEHS, NCI, USCG, NSF, NCIPC, and DOT, among others, interact with other agencies as partners in research, service, or dissemination
Global Partners	WHO, ILO, IARC, ISRP, WTO, and others, who collaborate with NIOSH as a trusted source for OSH research and service, data, and guidance
Insurance Organizations	Workers’ compensation, individual insurers, state agencies, and NGOs
Labor Organizations	AFL-CIO, Teamsters, North America’s Building Trade, SEIU, AFT, AFGE, and others who support NIOSH and its function for protection of worker health, safety, and well-being
Lobbyists	Support to increase congressional funding
Media	Involved in emergency response activities and were especially active during COVID-19 response
National Academies	Inform, shape, and assess the impact of the NIOSH Program Portfolio
OSH Professional Associations	Such as AIHA, ASSP, ACOEM, AAOHN, AOHP, APA, SOHP, NSC, SRA, and HFES
Standards Committees and Organizations	Brings NIOSH science to the standards
State and Local Agencies	Health departments and CSTE
U.S. Congress	Provides appropriations, including targeted funds, that support intramural and extramural research and service activities

**Table 3 ijerph-20-04333-t003:** Relevant history.

Time Period	Event
1970	Occupational Safety and Health (OSH) Act establishes NIOSH [44]
1994	NIOSH headquarters moves to Washington, DC, to improve partnership with DOL and increase presence on Capitol Hill
1995	OMB Paperwork Reduction Act governs the collection of information by federal agencies [45]
1996	NIOSH launches first 10-year partnership-driven research agenda (National Occupational Research Agenda [NORA]) [46]
1990s	Fewer regulations, increasingly diverse and older workforce, decreasing union membership
2001	Physical and cybersecurity programs increase as a result of 9/11 [47]
2000s	Big data era and emergence of electronic health records; AI and machine learning change the nature of work [48]
2011	World Trade Center Health Program established through the Zagroda Act [49]
2010s	Reemergence of gig economy and nonstandard work arrangements shifts responsibilities for risk from employer to employee [50]
2018	Revised Common Rule governs the terms and conditions of human subject research [51]
2019–2020	Significant malicious hacks of sensitive research data [52]

**Table 4 ijerph-20-04333-t004:** Driver names with descriptions, keywords from TIPPS, and count of supporting TIPPS.

Driver	Description	Keywords from TIPPS	Supporting TIPPS (*n*) *
Advanced Technology	Advances in data collection, automation, cloud computing, and artificial intelligence (AI) dramatically increase productivity and work customization but threaten to outpace the rates at which workers can be retrained, and systems can be built to cope with hazards. Human-machine interactions, both negative and positive, cause organizations to create new legal frameworks and modify policies and practices associated with worker privacy, engagement, and performance.	Advanced technology, Automation, Brain-machine interface, Cloud computing, Cybernetics, Electronic connectivity, Infrastructure upgrades, Neutral enhancement, Robots and AI, Pace of technology, Predictive modeling, Sustainable energy, Unintended consequences, Virtual environments	34
Climate and Energy	Growing evidence and concern around climate change are increasing interest in using alternative (renewable, clean) energy sources. Companies express plans to improve infrastructure and technology to move towards more sustainable and efficient processes (e.g., reduced carbon emissions and improved battery technology). These changes affect how OSH research and service are performed and create new hazards and risks for workers in multiple industries.	Carbon neutral, Climate, Energy, Sustainable architecture, Sustainable energy	5
Data Security	Use of new data collection and communications technologies leads to an increased need for cybersecurity, encryption, and worker data oversight. Joining up data from different sources is moving toward standardized interoperability methods to reduce time and effort in data collection. There is a growing need to ensure data security to protect the privacy of individuals and prevent cybercrime while leveraging the connectivity of data to improve health and safety.	Analytics, Big data, Data integration, Data security, Encryption, International agreements, Interoperability, Privacy and monitoring, Privacy rights, Tracking, Worker privacy	16
Knowledge Generation	Wavering trust in government as an information source challenges federal agencies striving to develop effective communication practices to counter misinformation. Funding and university-industry collaborations steer projects toward a broader benefit base, and information formats adapt to the demands of the communities they serve. Globalization and technological advances intensify international competition and investments in R&D.	International competition, Misinformation, Research funding, Research priorities, Research to practice	10
Social Credit	Social credit is a trustworthiness algorithm by which data on social standing rewards or punishes individual and corporate behavior. Data can be used to reward or punish behaviors at both the individual worker and organizational levels. The implementation of social credit scores impacts future employment, and employers can be held liable for employee actions. Consumers and workers actively seek to support companies that exhibit Corporate Social Responsibility to promote people, the planet, and profitability on equal terms.	Corporate Social Responsibility, Dehumanization of work, Environmental responsibility, Social credit, Social credit scores, Trust systems	5
Virtual Workplace	The boundaries of the workplace, including both laboratories and offices, are expanded by improved information communication technologies and new data channels. No longer limited to one physical location, workplaces are now ubiquitous—at home, on the road, in an airplane, or in an office. “Work” is defined by what you do, not where you go, each day.	New ways of working, Telework, Virtualization of the workplace, Workplace evolution	5
Work Arrangements	Nonstandard work arrangements have become the new normal. However, workers are challenged by these new arrangements and evolving hiring practices as employers seek skills over pedigree and accept more fluid employment histories. At the same time, the emergence of new industries is impacting work conditions. Known job risks and hazards may be eliminated or exacerbated as new challenges and opportunities are created.	Conditions of work, Gig economy, Talent market, Workplace, Work-life transformation	8
Workforce	Changes in U.S. population demographics influence labor market composition and employment patterns. Economic trends, individual preferences, and competing priorities delay retirement and extend the working-life continuum. Education systems are pressured to prepare future workers and help the current workforce upskill and reskill to meet growing talent needs. While physical safety remains a critical priority, employers must allocate resources to support all aspects of employee health and well-being.	Aging workforce, Diversity, Economics, Employee well-being, Future workforce, Labor markets and retirement, OSH workforce, Population, Psychosocial factors, Safety policy, Skill transformation, Total worker health, Universal basic income, Worker health, Worker well-being, Workforce demographics	33

* Total TIPPS reported = 116. Three Trends broadly discussed the future of work and could not be discretely mapped to a single driver.

**Table 5 ijerph-20-04333-t005:** Summary of the four future OSH scenarios.

Scenario Archetype	Scenario Title	Brief Description
Continuation	Boundaries Continue to Blur	The boundaries related to work locations, employment arrangements, work hours, the interface between work life and personal life, and human–machine interaction continue to blur.
Collapse	The Perfect Storm	Failure to adapt, coupled with a lack of trust and resources, forces people and organizations to rely on themselves to the detriment of worker health and safety.
New Equilibrium	Remote Controlled	Demands for new research on worker-centric arrangements, remote work, and human-machine collaborations strongly influence the allocation of OSH resources.
Transformation	One World Health	In this advanced tech world, mental health and data protections become central elements of an expanded OSH paradigm, research is driven by population need, and industries achieve one world health to sustain global workforce well-being.

**Table 6 ijerph-20-04333-t006:** Strategic focus areas for the future of OSH.

Strategic Focus Area	Underlying Strategic Issues
Data Security	The OSH workforce no longer meets worker needs due to worker fear of being monitored and data privacy issues (H2) Increased data security demands pose greater challenges to OSH research and surveillance activities (H2) Data security and privacy are an essential and unaddressed element of worker protection policies (H3)
Mental health	Significant increase in resources devoted towards the development of guidance for workplace psychosocial health and well-being policies (H1) New OSH competencies and a related discipline are needed to address significant worker mental health burden (H3)
Partnerships	OSH researchers lose the ability to access surveillance data and work sites due to changes in OSH policies and regulations (H2) Partner and key interest group connections must be built and maintained as virtual staff become more geographically dispersed (H2)
Research	Research into exposures to novel hazards and mental health requires reorganization and reeducation to incorporate expertise in new technologies (H1) Declining public and Congressional support limits mandated scope (H2) OSH community must continue to address traditional hazards as new OSH issues emerge at a rapid pace and require attention (H2) OSH research is driven by new OSHQ metrics and the needs of priority populations and key interest groups, requiring a new approach to the OSH research portfolio (H3)
Virtual Work	Federal human resources policies for remote work are not in line with private industry (H1)

**Table 7 ijerph-20-04333-t007:** Strategic options and actions.

**Data Security**	
Mid-term	Improve communication and synergism between data security and science personnel. Provide a clear understanding of roles and responsibilities regarding what IT security and data analyses services can be provided centrally versus what individual researchers can and should do. Create data security systems that respond to changing scientific needs and enhance OSH work within CDC, with other federal agencies, and with external partners.
Far-term	Develop new data security and privacy paradigms for OSH research. Recruit IT and research resources to meet the data security demands of the future portfolio. Implement a new paradigm to meet data security and privacy needs that support OSH research.
**Mental Health**	
Near-term	Acquire human capital resources with requisite knowledge and competencies (hiring, IPAs, contracting, etc.). Increase occupational mental health surveillance efforts. Establish workgroups to identify existing mental health literature to inform policy perspectives and identify research gaps. Develop protocols for addressing mental health concerns through holistic approaches that include family and community considerations. Conduct formative research to identify how to establish trust and communication about mental health challenges in the workplace.
Far-term	Establish new working groups devoted to exploring and understanding worker mental health. Recruit OSH scientists and researchers with related skill sets. Integrate mental health priorities into OSH research portfolios. Redesign OSH research and service activities to align with new OSHQ metrics and the needs of priority populations and key interest groups.
**Partnerships**	
Mid-term	Nurture relationships with longstanding partners and key interest groups while simultaneously casting a broader net for new connections. Embed OSH staff into different industries to strategically expand expertise and connections. Consider alternative organizations and structures to increase opportunities that establish and strengthen relationships with partners and stakeholders.
**Research**	
Near-term	Acquire human capital resources with requisite knowledge and competencies (hiring, IPAs, contracting, etc.). Identify high-priority industry sectors and early adopters within those sectors. Collaborate with early adopters in high-priority sectors to initiate exploratory studies of the effects of new technologies. Pursue intramural and extramural partnerships to document early trends and establish best practices to influence general practices and culture.
Mid-term	Ensure intramural and extramural research adequately focuses on traditional hazards, critical emerging issues, and new health and safety outcomes. Improve alignment and transparency between intramural and extramural research communities to ensure comprehensive attention is given to traditional and emerging health and safety issues and outcomes. Enhance systems that track traditional hazards and aid in the identification of emerging issues across industry sectors. Ensure the OSH workforce includes the expertise needed to address traditional hazards and emerging issues across all major industry sectors.
Far-term	Redesign OSH research and service activities to align with new OSHQ metrics and the needs of priority populations and key interest groups.
**Virtual Work**	
Near-term	Modernize and clarify HR policies related to remote work. Re-evaluate the time-based boundaries on workdays.

## Data Availability

Not applicable.

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
