# Peer review of "Four Futures for Occupational Safety and Health"

_ijerph, 2023, doi:10.3390/ijerph20054333_

Round 1

Reviewer 1 Report

Dear Authors,

This paper describes the inaugural NIOSH strategic foresight project, which was designed to promote institutional capacity in strategic foresight and improve preparedness by considering how the future will impact occupational safety and health (OSH) research and service activities. With multidisciplinary teams of subject matter experts at NIOSH, the authors undertook extensive exploration and information synthesis to inform the development of four alternative future scenarios for OSH. They briefly describe these futures and discuss their implications for OSH. They identify strategic responses that can serve as the basis for an action-oriented roadmap toward a preferred future, too.

The article is remarkably interesting and well developed. However, slight revisions are necessary:

1- A scientific text should be written in the third person avoiding we. Please, to improve the paper, eliminate the use of the first-person plural in the text.

2-  These two sentences (Lines 187-190) can be merged into one:

With the drivers defined, we constructed a cross-impact matrix, which is an analytical technique used in foresight to explore how drivers might interact in the future [24,142-144].”

“The goal of the cross-impact matrix is to explore how different drivers may interact in the future [144-146].”

3-   In the sentences:

“The title, abstract, and key drivers for each of the final four scenarios are provided below” (Lines 237-238) and “All four scenarios are described in more detail in the following section” (Lines 241-242),

it is better to identify numerically the sections to which they refer, rather than using “in the following section” or “… are provided below”

   4-  In table 6, under Mental Health, there are two Far-term groupings,                 does the first one refer to Mid-term? Clarify this duplication.

Sincerely

Author Response

Please see the attachment for a detailed response to each of Reviewer 1 comments. Thank you.

Reviewer 2 Report

This paper describes NIOSH's framework and ignited strategies more than it is a review of scientific value. In addition, the description is very long, a lot of information in the tables is repeated in the text, some figures could be integrated (e.g. fig. 5 and 6). The limitation of this work is also presenting observations related to US population and conditions. 

Author Response

Please see the attachment for a detailed response to each of Reviewer 2 comments.  Thank you.

Reviewer 3 Report

Dear authors, the manuscript looks more like a manual than a review article. Since it is a review article, the review must be well-founded and the methodology must be very detailed. What databases are used, how many articles are used for the review and how many are excluded. Most of the references are web pages, which is not appropriate for an article of these characteristics.

The introduction is not clear enough and does not cover the current situation of OSH. It should be completed with recent references and at least mention OSH5.0.

How are the stakeholders in Table 2 defined?

How are the 4 scenarios or archetypes defined?

Line 79, the web address should not appear in the title of the figure, but the citation to the reference, as has been done in Figure 3.

Is Figure 2 your own elaboration? If not, cite the source.

It is not clear how the matrix in Figure 4 is constructed.

In general the document is not very well understood at the global level, it seems that they are parts of a project that have been done but are difficult to connect.

The language used throughout the document should be inclusive, so the document should be revised.

Author Response

Please see the attachment for the detailed response to each of Reviewer 3 comments.  Thank you.

Reviewer 4 Report

Interesting paper which presents four scenarios of plausible futures for OSH based on identified key drivers of change and which proposes key strategic areas on which the focus should be placed for the future. The paper is well-structured and documented and the approach taken to develop the scenarios seems to rely on a framework and model used  before in foresight research.

While the paper presents a good synthesis of the various results that have been obtained, it perhaps fails to be explicit enough on how some of the results have been generated and what was the role of the foresight team participants (or stakeholders or others?) in generating those results. This is particularly true as it relates to the scoring that was attributed in Figure 4 to express the cross-impact of the drivers of change, and to the identification of the strategic focus areas and issues appearing in Table 5 and to the list of strategic actions identified in Table 6. Was there a consensual or some other approach taken? These results clearly need some justification within the paper.

Finally, considering that the paper is intended to be published in an International Journal, it should perhaps be indicated in section 5 if the recommendations listed in Table 6 apply more specifically to OSH work within CDC or if generalization elsewhere is also a possibility.

Author Response

Please see the attachment for the detailed response to each of Reviewer 4 comments. Thank you.

Round 2

Reviewer 3 Report

Changes are accepted for publication